# Navigation of a Differential Wheeled Robot Based on a Type-2 Fuzzy Inference Tree

**Dante Mújica-Vargas** [1,*] , **Viridiana Vela-Rincón** [1] , **Antonio Luna-Álvarez** [1] , **Arturo Rendón-Castro** [1] , **Manuel Matuz-Cruz** [2] and **José Rubio** [3]

[1] Department of Computer Science, Tecnológico Nacional de México/CENIDET, Interior Internado Palmira S/N, Palmira, Cuernavaca 62490, Mexico

[2] Departamento de Sistemas Computacionales, Tecnológico Nacional de México/ITTapachula, Tapachula Chiapas 30700, Mexico

[3] Sección de Estudios de Posgrado e Investigación, ESIME Azcapotzalco, Instituto Politécnico Nacional, Av. de las Granjas No. 682, Col. Santa Catarina, Ciudad de México 02250, Mexico

* Correspondence: dante.mv@cenidet.tecnm.mx

**Abstract:** This paper presents a type-2 fuzzy inference tree designed for a differential wheeled mobile robot that navigates in indoor environments. The proposal consists of a controller designed for obstacle avoidance, a controller for path recovery and goal reaching, and a third controller for the real-time selection of behaviors. The system takes as inputs the information provided for a 2D laser range scanner, i.e., the distance of nearby objects to the robot, as well as the robot position in space, calculated from mechanical odometry. The real performance is evaluated through metrics such as clearance, path smoothness, path length, travel time and success rate. The experimental results allow us to demonstrate an appropriate performance of our proposal for the navigation task, with a higher efficiency than the reference methods taken from the state of the art.

**Keywords:** indoor navigation; differential wheeled mobile robot; type-2 fuzzy inference tree; 2D laser range scanner; mechanical odometry; ROS

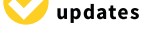



## 1. Introduction

Robot navigation is the problem of guiding a robot toward a goal [1]. The autonomous navigation is conditioned to the planning of motions, i.e., the interrelationship between tasks, such as the environment perception, path planning, and path tracking, among others. Thus, it might be said that the planning of motions plays a key role for mobile robotic systems. In mobile robotics, the principle behind the motion planning problem is to guide a robot through an environment and interact with objects or tools, avoiding obstacles and taking into account any restrictions, such as those of mobility and energy [2]. Motion planning for mobile robots is one of the most challenging problems; over time, its different inherent subproblems have been expanded by adding variables such as dynamic obstacles, complex environments, dead reckoning, linear and angular velocities control, holonomic and nonholonomic constraints, energy, fault tolerance, algorithms runtime and so on.

Motion planning and its inherent problems have been a researched area for several years. The state of the art suggests different investigation lines; just to name a few, the following may be mentioned. For the shortest path planning, there are planners, such as Dijkstra [3], A* [4], and Theta* [5], and based on them, there are so important proposals [6–13]. For path planning, there exist also developments based on optimal control theory [14,15]. For motion planning are highlighted such methods as Hybrid A* (HybridAStar) [16], rapidly exploring random tree (RRT) [17], bidirectional rapidly exploring random tree (BiRRT) [18] and probabilistic road map (PRM) [19]. In terms of path tracking, i.e., where a reference trajectory is known beforehand, there are approaches based on predictive control [20], based on fuzzy theory [21], and based on classic control [22]. There

is also a significant contribution related with the obstacles avoidance and the reaching goal time reduction; the broad range includes the A* algorithm [23,24], neural networks [25], genetic algorithms [26,27], fuzzy logic [28], particle swarm optimization [29], and ant colony optimization [30], among others [31,32].

There is an abundance of literature on this subject to cover in the current study; notwithstanding, it is necessary to mention merged approaches. Some developments may be cited, for instance, in [33], an ant colony algorithm was proposed using an evaluation function of the A* algorithm to accelerate the algorithm convergence rate. It was designed for the path planning. In this regard, two ant colony algorithms were executed independently and successively in [34]. On the other hand, ant colony was fused with fuzzy logic in [35]. In [36], the shortest path problem was addressed with a PRM in cooperation with a fuzzy controller; this last helped in having smooth curves by adjusting the turning angles and correcting the position and orientation of the robot. A knowledge-based neural fuzzy controller was introduced in [37], and was evaluated using a PIONEER 3-DX platform for navigation and obstacle avoidance tasks. Finally, there are a few contributions where the technology and new computational paradigms are considered. By this way, in [38], the robot operating system (ROS) and sensors such as 2D LiDAR and RGB-D, were used to perform the navigation task with dynamic obstacle avoidance capability. In [23], the pure pursuit algorithm was improved by considering the 2D LiDAR point-cloud features with a deep convolutional network. LiDAR sensors have also been combined with deep convolutional neural network models to navigation control for wheeled mobile robots, as described in [39].

In this study, we concentrate on the motion planning, by providing autonomy to a differential wheeled robot. That is to say, the mobile robot navigates and interacts with objects within the environment, and intrinsically it is able to know where it is located and where it will go. The contributions of this study are summarized as follows:

1. A type-2 fuzzy inference tree is proposed as real-time motion planning in the mobile robot navigation problem.
2. Type-2 fuzzy theory is the theoretical support of the current system, which makes it different from common fuzzy controllers used in mobile robotics, where type-1 fuzzy theory is used.
3. The tree scheme is suggested as a solution to reduce the computational complexity, by the delegation of tasks, specifically, obstacle avoidance, path recovering, and goal reaching, as well as behavior coordination.
4. The proposal is fully functional on physical robotic platforms, in contrast with many state-of-the-art approaches that are computational simulations.
5. The performance of the proposed system was evaluated on a real environment with a smooth floor and rough surfaces.

The rest of this paper is organized as follows. In Section 2, the system composition is described in detail, with a special emphasis on the proposed type-2 fuzzy inference tree and its mathematical background. Experimental results and a comparative analysis with other current methods in the literature are presented in Section 3. The concluding section gives a synopsis of the principal results and recommendations for future work.

## 2. Materials and Methods

### 2.1. System Composition

The system used in this study is shown in Figure 1. It consists of a Roomba 620 mobile robot with two differential drive wheels fitted with incremental wheel encoders; its linear $v$ and angular $w$ velocity limits comprise $-0.5 \leq v \leq 0.5$ (m/s) and $-4.25 \leq w \leq 4.25$ (rad/s). It also includes a 360° 2D laser range scanner RPLIDAR-A3 with a maximum ranging distance of 25 m, sampling frequency of 20 Hz and angular resolution of 0.25°. These devices are controlled by a local ARM CPU set up with ROS Melodic; in particular, it develops three tasks. The first activity is setting up both linear velocity $v$ and steering $w$, as well as taking measures of angular velocities for right and left wheels ($w_r$ and $w_l$,

respectively). This information is used to compute the robot position $P = \{x, y, \theta\}$ over time, by means of odometry kinematics. The second task implies detecting nearby objects; to this end, distance $r$ and angle $\phi$ readings are taken, using a laser scanner. The sensor is able to acquire $360/0.25 = 1440$ measures, every $1/20 = 0.05$ seconds; for simplicity, the information is represented in matrix form as $S = \{r_i, \phi_i\}$. The latest process involves transmitting $P$ and $S$ arrays over Wi-Fi to the remote CPU, just as receiving velocity and steering commands. On the remote desktop computer, provided with Matlab, Fuzzy Logic and ROS toolboxes, the proposed control system is executed in real time.

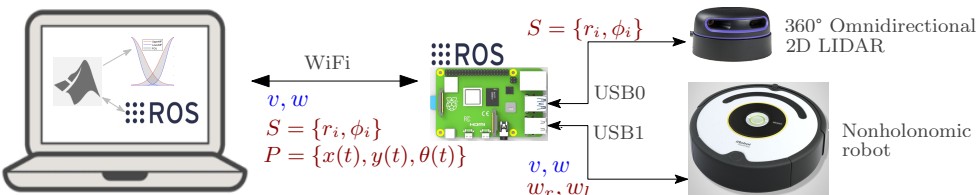

**Figure 1.** Composition of used system.

### 2.2. Laser Range Scanner Measures

The LIDAR sensor scans the environment in an omnidirectional way, with an angle resolution of $0.25°$, plus a 25 m maximum operating range. For appropriate sensor operation within this framework research, the next constraints are set as: (a) Just $180°$ frontal coverage is required. (b) A detection range between 0.15 m and 0.55 m is considered since the robot navigates in indoor environments. In $180°$ coverage, it is possible to acquire $180/0.25 = 720$ measures; these readings cannot be considered inputs to the fuzzy tree system. To address this issue, it is suggested to work with $30°$ regions, as shown in Figure 2. For simplicity, they are labeled as Right-Down (RD), Right-Up (RU), Right-Front (RF), Left-Front (LF), Left-Up (LU) and Left-Down (LD).

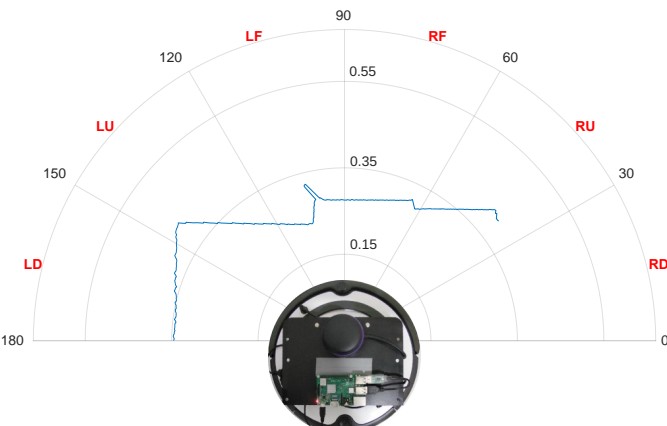

**Figure 2.** LIDAR coverage.

For every zone, $30/0.25 = 120$ measures ($m$) may be acquired. Readings with the minimum values are of special interest, since they allow locating the closest objects. They are computed as follows:

$$\text{RD}_{min} = \min(m_{0.0}, m_{0.25}, \cdots, m_{30.0}), \tag{1}$$

$$\text{RU}_{min} = \min(m_{30.25}, m_{30.5}, \cdots, m_{60.0}), \tag{2}$$

$$\text{RF}_{min} = \min(m_{60.25}, m_{60.5}, \cdots, m_{90.0}), \tag{3}$$

$$\text{LF}_{min} = \min(m_{90.25}, m_{90.5}, \cdots, m_{120.0}), \tag{4}$$

$$\text{LU}_{min} = \min(m_{120.25}, m_{120.5}, \cdots, m_{150.0}), \tag{5}$$

$$\text{LD}_{min} = \min(m_{150.25}, m_{150.5}, \cdots, m_{180.0}). \tag{6}$$

### 2.3. Odometry

The simplest and commonly used approach for robot localization is based on mechanical odometry; it uses information from wheel encoders to estimate all changes in position over time (see Figure 3).

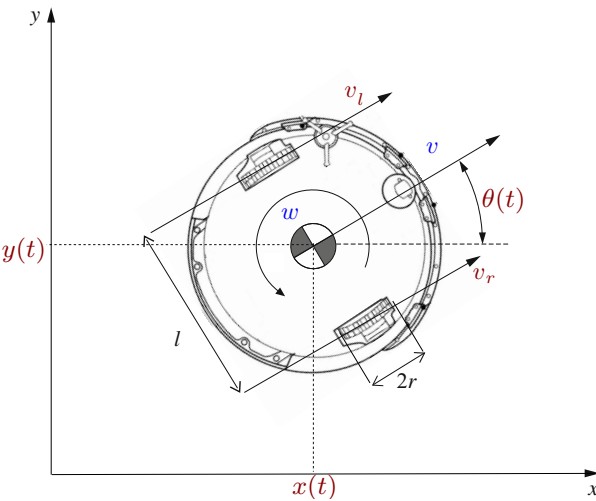

**Figure 3.** Differential robot odometry.

To understand the differential robot motion, it is necessary to compute how much every wheel contributes to robot velocity, by considering the following expressions [40]:

$$v_r = 2\pi r \cdot \frac{n_r(t) - n_r(t - \Delta t)}{N \Delta t}, \tag{7}$$

$$v_l = 2\pi r \cdot \frac{n_l(t) - n_l(t - \Delta t)}{N \Delta t}, \tag{8}$$

where $v_r$ and $v_l$ are the right and left wheel velocities, respectively, $r$ is the wheel radius, $N$ is the encoder ticks number for one full wheel revolution, $n_r$ and $n_l$ are encoder ticks of the right and left wheel at time $t$, respectively, and $n_r(t - \Delta t)$ and $n_l(t - \Delta t)$ are the same quantities at the previous sampling time. Expressions (7) and (8) let us compute both linear $v$ and angular $w$ velocities of the robot as

$$v = \frac{v_r + v_l}{2}, \tag{9}$$

$$w = \frac{v_r - v_l}{l}, \tag{10}$$

where $l$ stands for the distance between the wheels. The robot position ($x(t)$ and $y(t)$), as well as the orientation ($\theta(t)$) can then be estimated with discrete numerical integration (e.g., Euler) as

$$x(t) = x(t - \Delta t) + v\cos(\theta(t))\Delta t, \tag{11}$$

$$y(t) = y(t - \Delta t) + v\sin(\theta(t))\Delta t, \tag{12}$$

$$\theta(t) = \theta(t - \Delta t) + w\Delta t. \tag{13}$$

In particular for Roomba 620 robot, the counts per revolution are $N = 508.8$, the wheel radius is $r = 0.036$ (m) and the wheel baseline is $l = 0.235$ (m).

### 2.4. Proposed Type-2 Fuzzy Inference Tree

The type-2 fuzzy inference tree proposed in this study is shown in Figure 4. In brief, the behaviors controller takes all inputs to analyze the environment and the robot position with respect to the goal. If there are obstacles near the robot, the controller selects the obstacle avoidance behavior; otherwise, the goal reaching behavior is chosen.

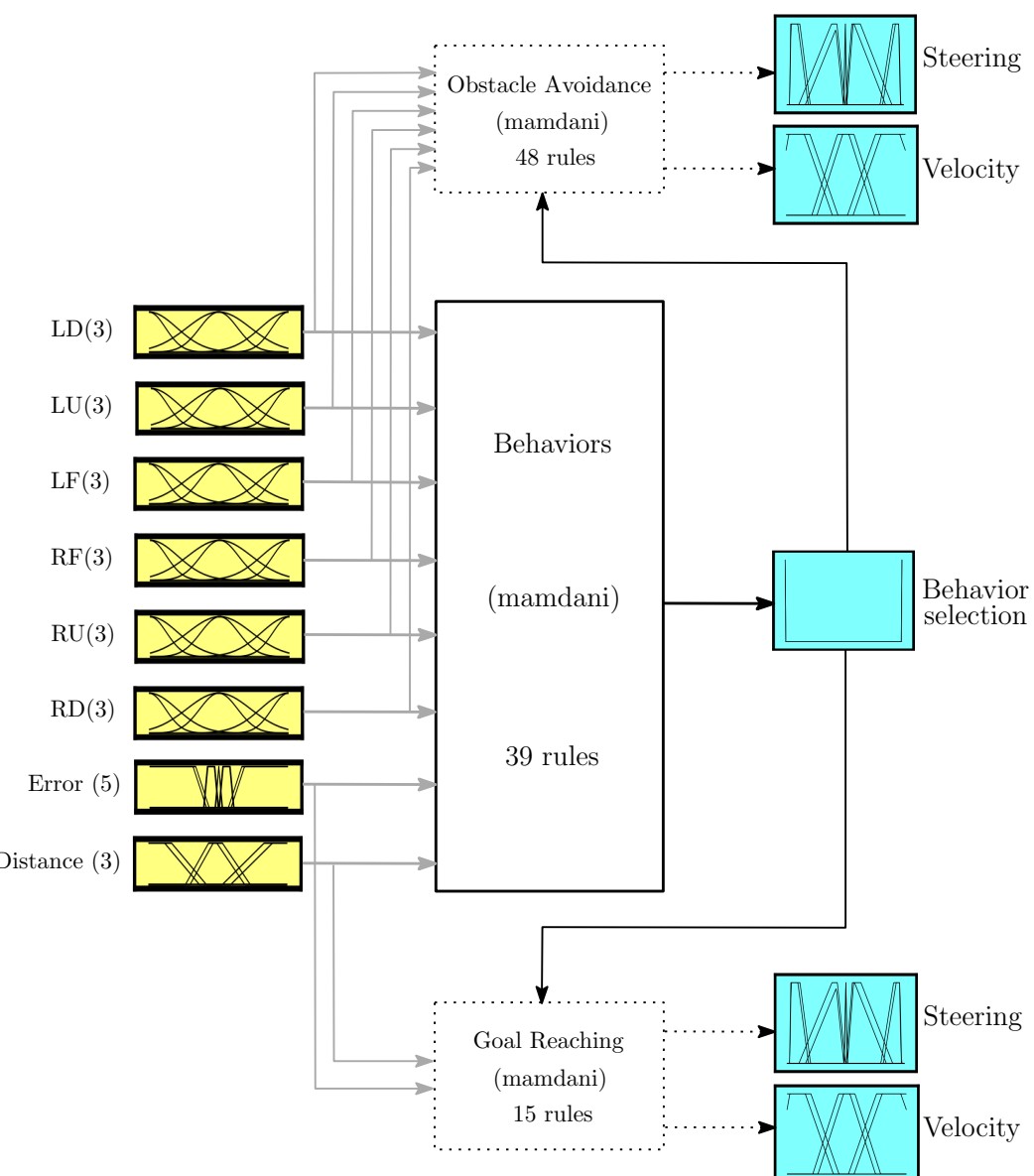

**Figure 4.** Proposed type-2 fuzzy inference tree.

Figure 5 depicts the type-2 fuzzy sets for all inputs, by convention they can be considered two subsets. RD, RU, RF, LF, LU and LD variables are the first subset; this allows us to quantify the obstacles proximity during the navigation. Error and distance variables are the second subset; it enables us to measure the error angle with respect to goal and the distance to reach the target. The operating range for each variable was established considering that the robot navigates only in indoor environments; however, it can be extended to outdoor environments.

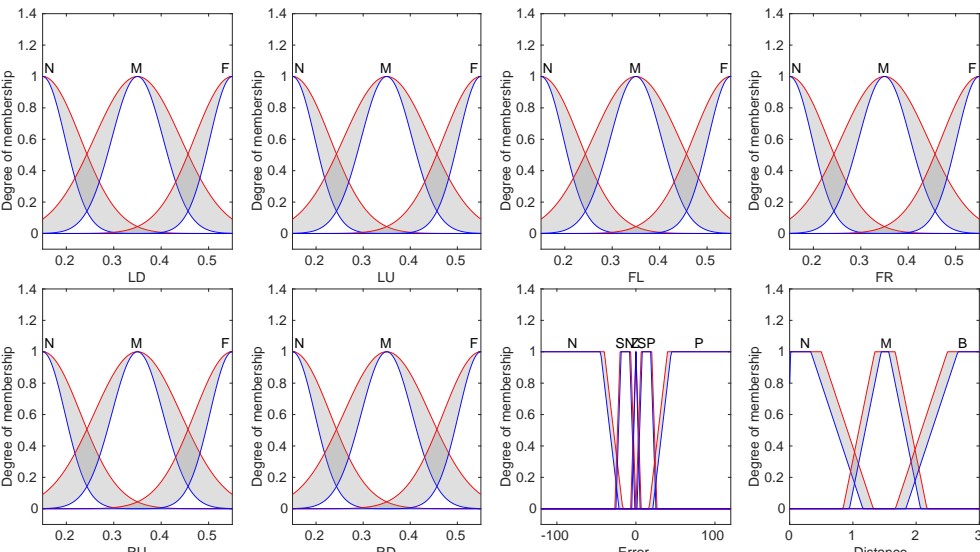

**Figure 5.** Type-2 fuzzy sets in the inputs.

All variables of the first subset are modeled by three Gaussian fuzzy sets, denoted as Near (N), Medium (M) and Far (F). For the second one, they are modeled independently, i.e., the error variable is modeled with five trapezoidal fuzzy sets named Negative (N), Small Negative (SN), Zero (Z), Small Positive (SP) and Positive (P). By contrast, the distance variable is described with three trapezoidal fuzzy sets, so-called Near (N), Medium (M) and Far (F).

Figure 6 depicts the output variables; to achieve a better understanding, also consult Figure 4. At first instance, the behaviors output is modeled by two sets titled GoalReaching and ObsAvoidance; this output develops a binary classification or a switching between behaviors. On the other hand, both obstacle avoidance and goal reaching controllers have the same outputs, which allow to adjust the steering and velocity. For this purpose, the steering output is modeled by five trapezoidal fuzzy sets labeled as Right (R), Right Front (RF), Front (F), Left Front (LF) and Left (L). This variable is the angular velocity; in this study it is considered a range of $-1 \leq w \leq 1$ radians per second. Finally, Velocity variable stands for the linear velocity, it is considered $0 \leq v \leq 0.2$ m per second. The output is modeled by three sets, Zero (Z), Small Positive (SP) and Positive (P). Negative velocities are not considered, since backward movements are not required.

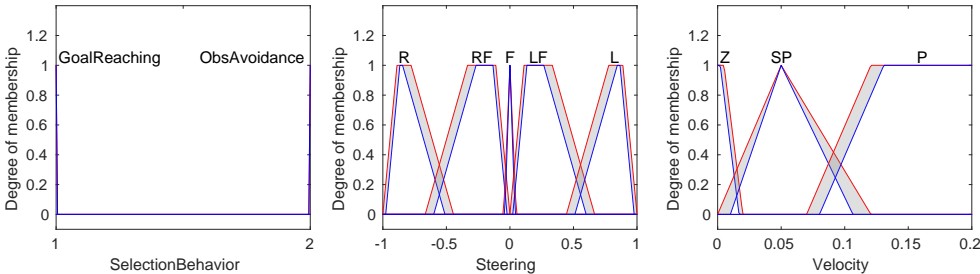

**Figure 6.** Type-2 Fuzzy Sets in the Outputs.

*2.5. Mathematical Support of the Proposal*

The current proposal consists of three multiple-input multiple-output type-2 fuzzy controllers. In order to simplify its mathematical support, it can be assumed that each system can be seen as a combination of multiple-input single-output type-2 fuzzy controllers. A conventional type-2 fuzzy system with a type-reduction + defuzzification contains five components which are the fuzzifier, rule base, fuzzy inference engine, type-reducer and

defuzzifier [41]. To describe these components, the following conceptualization can be standardized and applicable for all three controllers.

### 2.5.1. Fuzzifier

Fuzzifier takes a crisp input vector with $n$ samples provided by the sensors $(x_1, \cdots, x_n)^T \in X_1 \times \cdots \times X_n \equiv \mathbf{X}$ and maps it into type-2 fuzzy sets. The literature suggests type-2 singleton, type-1 non-singleton and type-2 non-singleton fuzzifiers [41,42]. In this study, type-2 singleton was chosen since it is fast to compute and thus suitable for the robot real-time operation. Given a type-2 fuzzy set $\tilde{X}_i$, a type-2 singleton fuzzifier is one for which $\mu_{\tilde{X}_i}(x_i|x_i') = 1/1$ when $x_i = x_i'$ and $\mu_{\tilde{X}_i}(x_i|x_i') = 1/0$ if $x_i \neq x_i'$.

### 2.5.2. Rule Base

In [41], it was stated that the rules for a type-2 fuzzy controller were similar as type-1 fuzzy controller, provided that the type-1 fuzzy sets were replaced by type-2 fuzzy sets. Therefore, the structure of the $k$-th rule for a multiple-input single-output type-2 fuzzy controller based on Mamdani inference is given as

$$\tilde{R}^k : \text{IF } x_1 \text{ is } \tilde{F}_1^k \text{ and } \cdots \text{ and } x_n \text{ is } \tilde{F}_n^k \text{ THEN } y \text{ is } \tilde{G}^k, \tag{14}$$

where $\tilde{F}_1^k \in \{\tilde{X}_{1j}\}_{j=1}^{Q_1}, \cdots, \tilde{F}_n^k \in \{\tilde{X}_{nj}\}_{j=1}^{Q_n}$ stand for the $Q_i$ linguistic terms that modeled each $i$-th input. Likewise, $\tilde{G}^k \in \{\tilde{Y}_j\}_{j=1}^{Q}$ describes the linguistic terms that model the output.

### 2.5.3. Fuzzy Inference Engine

The inference engine combines rules and makes a mapping from input type-2 sets to output type-2 sets [41]. For this purpose, a set of $1 \leq k \leq M$ rules having $n$ inputs $x_1 \in X_1, \cdots, x_n \in X_n$, as well as an output $y \in Y$ are taken into account, then the transformation process is formalized as

$$\tilde{R}^k : \tilde{F}_1^k \times \cdots \times \tilde{F}_n^k \mapsto \tilde{G}^k = \tilde{A}^k \mapsto \tilde{G}^k, \tag{15}$$

where $\tilde{A}^k = \tilde{F}_1^k \times \cdots \times \tilde{F}_n^k$. $\tilde{R}^k$ can be described by the membership function $\mu_{\tilde{R}^k}(\mathbf{x}, y) = \mu_{\tilde{R}^k}(x_1, \cdots, x_n, y)$, which implies that:

$$\mu_{\tilde{R}^k}(\mathbf{x}, y) = \mu_{\tilde{A}^k \mapsto \tilde{G}^k}(\mathbf{x}, y), \tag{16}$$

$\mathbf{x}$ stands for a condensed representation of the $n$ inputs. The expansion of (16) is given by

$$\mu_{\tilde{A}^k \mapsto \tilde{G}^k}(\mathbf{x}, y) = \mu_{(\tilde{F}_1^k \times \cdots \times \tilde{F}_n^k)}(\mathbf{x}) \sqcap \tilde{G}^k = \left[\sqcap_{i=1}^n \mu_{\tilde{F}_i^k}(x_i)\right] \sqcap \mu_{\tilde{G}^k}(y), \tag{17}$$

$\sqcap$ denotes the intersection of the type-2 fuzzy sets. Moreover, the $n$-dimensional input to the rule set is given by type-2 fuzzy set $\tilde{A}_{\mathbf{x}'}$ whose membership function is stated as

$$\mu_{\tilde{A}_{\mathbf{x}'}}(\mathbf{x}) = \mu_{\tilde{X}_1(x_1|x_1')} \sqcap \cdots \sqcap \mu_{\tilde{X}_n(x_n|x_n')} = \sqcap_{i=1}^n \mu_{\tilde{X}_i(x_i|x_i')}. \tag{18}$$

To establish a direct connection between the input and output, it should be kept in mind that rules determine which sets are triggered on the output. This is formalized by

$$\mu_{\tilde{B}^k}(y, \mathbf{x}') = \sqcup_{x \in X}\left[\mu_{\tilde{A}_{\mathbf{x}'}}(\mathbf{x}) \sqcap \mu_{\tilde{R}^k}(\mathbf{x}, y)\right]. \tag{19}$$

If (17) and (18) are substituted into (19), an input–output relation can be expressed as

$$\mu_{\tilde{B}^k}(y, \mathbf{x}') = \left\{\sqcup_{i=1}^n\left[\sqcup_{x_i \in X_i}\left(\mu_{\tilde{X}_i(x_i|x_i')} \sqcap \mu_{\tilde{F}_i^k(x_i)}\right)\right]\right\} \sqcap \mu_{\tilde{G}^k(y)}. \tag{20}$$

The bracket term is so-called a firing set [41], it is denoted as $F^k(\mathbf{x}')$, and it allows to simplify (20) in the following way:

$$\mu_{\tilde{B}^k}(y, \mathbf{x}') = F^k(\mathbf{x}') \sqcap \mu_{\tilde{G}^k(y)}, \tag{21}$$

$\mu_{\tilde{B}^k}(y, \mathbf{x}')$ stands for the membership function for a fired rule output set. For a type-2 Mamdani fuzzy inference system, the singleton fuzzification [41] is rewritten as

$$\mu_{\tilde{B}^k}(y, \mathbf{x}') = \left(1 / \left[\underline{f}^k(\mathbf{x}'), \overline{f}^k(\mathbf{x}')\right]\right) \sqcap \mu_{\tilde{G}^k(y)}, \tag{22}$$

where $\underline{f}^k(\mathbf{x}')$ and $\overline{f}^k(\mathbf{x}')$ depict the lower and upper membership degrees of the footprint of uncertainty, given by

$$\underline{f}^k(\mathbf{x}') = \underline{\mu}_{\tilde{F}_1^k}(x_1') \star \underline{\mu}_{\tilde{F}_2^k}(x_2') \star \cdots \star \underline{\mu}_{\tilde{F}_n^k}(x_n'), \tag{23}$$

$$\overline{f}^k(\mathbf{x}') = \overline{\mu}_{\tilde{F}_1^k}(x_1') \star \overline{\mu}_{\tilde{F}_2^k}(x_2') \star \cdots \star \overline{\mu}_{\tilde{F}_n^k}(x_n'), \tag{24}$$

where $\star$ is used to denote the minimum or product implication method.

### 2.5.4. Type-Reduction

When a type-2 fuzzy inference system is used in a real-world application, the issue of the computational cost and complexity become non-trivial [41]. In this regard, a type-reduction process can be performed on $\mu_{\tilde{B}^k}(y, \mathbf{x}')$ in order compute an output. Output region $\mathcal{Z}_{\tilde{B}}$ requires the entire footprint of uncertainty of $\tilde{B}$ or a sampled version of it as its starting point, and is

$$\mathcal{Z}_{\tilde{B}} = 1 / [z_l(\tilde{B}), z_r(\tilde{B})], \tag{25}$$

where $z_l(\tilde{B})$ and $z_r(\tilde{B})$ are the left and right transition of the membership function that defines the output region. They can be computed by means of

$$z_l(\tilde{B}) = \frac{\sum_{i=1}^{\mathcal{L}} y_i \overline{\mu}_{\tilde{B}}(y_i | \mathbf{x}') + \sum_{i=\mathcal{L}+1}^{\mathcal{N}} y_i \underline{\mu}_{\tilde{B}}(y_i | \mathbf{x}')}{\sum_{i=1}^{\mathcal{L}} \overline{\mu}_{\tilde{B}}(y_i | \mathbf{x}') + \sum_{i=\mathcal{L}+1}^{\mathcal{N}} \underline{\mu}_{\tilde{B}}(y_i | \mathbf{x}')}, \tag{26}$$

$$z_r(\tilde{B}) = \frac{\sum_{i=1}^{\mathcal{R}} y_i \overline{\mu}_{\tilde{B}}(y_i | \mathbf{x}') + \sum_{i=\mathcal{R}+1}^{\mathcal{N}} y_i \underline{\mu}_{\tilde{B}}(y_i | \mathbf{x}')}{\sum_{i=1}^{\mathcal{L}} \overline{\mu}_{\tilde{B}}(y_i | \mathbf{x}') + \sum_{i=\mathcal{R}+1}^{\mathcal{N}} \underline{\mu}_{\tilde{B}}(y_i | \mathbf{x}')}. \tag{27}$$

If the centroid approach is considered, a continuous output region is generated from the activated rules; therefore, sample values of $\underline{\mu}_{\tilde{B}}(y_i | \mathbf{x}')$ and $\overline{\mu}_{\tilde{B}}(y_i | \mathbf{x}')$ can be computed. In this study, the enhanced Karnik and Mendel (EKM) algorithm [41] was chosen, largely because it has a better initialization, which reduces the number of iterations, as well as the fact that it has a subtle computing technique, which is used to reduce the computational cost of each algorithm iteration, among other aspects.

### 2.5.5. Defuzzification

The last process in a fuzzy system implies the defuzzification. Taking into account $[z_l(\tilde{B}), z_r(\tilde{B})]$, the defuzzified output $y_{\mathcal{Z}}(\mathbf{x}')$ for a type-2 Mamdani fuzzy inference system may be computed via

$$y_{\mathcal{Z}}(\mathbf{x}') = \frac{z_l(\tilde{B}) + z_r(\tilde{B})}{2}. \tag{28}$$

## 3. Experimental Results

To assess and confirm the current proposal performance, two experiments were carried out on an indoor environment, with no dynamic objects. For the first trial, an obstacle-free environment was considered, while for the second one, some objects were included to make

it difficult to navigate and reach the goal. As seen in Figure 7, the hallways of the Computer Science Department building of CENIDET (National Center of Research and Technological Development, Spanish acronym) constitute our experimentation environment.

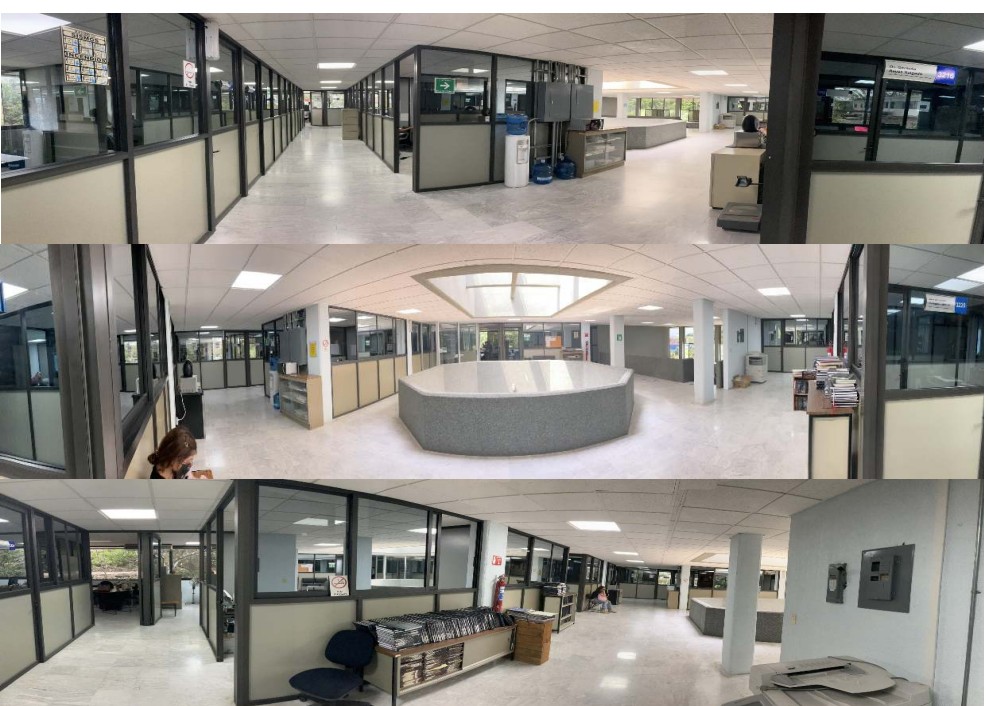

**Figure 7.** Real navigation environment.

Figure 8 depicts sketches of the real dimensions, as well as an occupancy grid map built from lidar scans and poses collected by the robot through a remotely controlled navigation. All unnecessary objects were removed from the building structure to have a completely obstacle-free map.

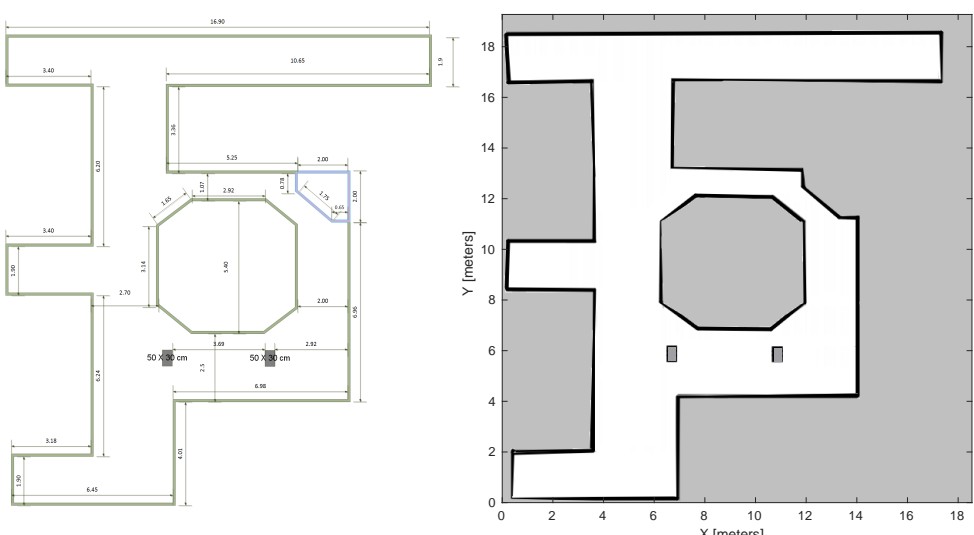

**Figure 8.** Real measurements and occupation map of the work environment.

For the purpose of having a performance benchmark, the current proposal and some well-known methods from the literature were evaluated to perform the 2D path planning and avoiding obstacles for a differential drive robot. In particular, the comparative methods were HybridAStar [16], RRT [17], BiRRT [18] and PRM [19]. All of them are available in

Matlab; therefore, they were not programmed manually, not to impair their performance. In this regard, these are easily workable with a robotic platform in a physical way. It is necessary to clarify that although novel techniques for trajectory tracking have recently been presented in the literature, they cannot be used in the context of this research work. The aim is not just to follow a path, since the avoidance of obstacles and the recovery of the trajectory must be done, or failing that, new trajectories must be calculated and followed during navigation.

For an objective comparison, some performance metrics [2,43] were taken as a reference, such as clearance, path smoothness, path length, travel time and success rate.

**Clearance:** this metric is related to the distance from the trajectory points to the closest obstacle, and it is defined as

$$\zeta = \frac{1}{n}\sum_{i=1}^{n}\delta_i = \frac{1}{n}\sum_{i=1}^{n}\sqrt{(x_i - x_{closeObst})^2 + (y_i - y_{closeObst})^2}, \tag{29}$$

where $\delta_i$ represents the Euclidean distance from the $i$-point of the path to the closest obstacle and $n$ is the number of points of the path. A higher clearance value indicates a lower risk of collision.

**Path smoothness:** the smoothness of a path refers to the amplitude of the angles that are described while the robot follows the trajectory

$$\kappa = \frac{1}{n}\sum_{i=2}^{n}\alpha_i = \frac{1}{n}\sum_{i=2}^{n}\left|\arctan\left(\frac{y_{i+1} - y_i}{x_{i+1} - x_i}\right) - \arctan\left(\frac{y_i - y_{i-1}}{x_i - x_{i-1}}\right)\right|, \tag{30}$$

where $\alpha_i$ stands for the angle between two consecutive segments of a path with $n$ segments. A lower smoothness value indicates smoother path.

**Path length:** is defined as the sum of the distances from one way point to the next one in the planner state space.

$$\ell = \sum_{i=1}^{n-1}\sqrt{(x_i - x_{i+1})^2 + (y_i - y_{i+1})^2}. \tag{31}$$

**Travel time:** is the time span that the robot travels the entire path from the start to the goal.
**Success rate:** is equal to the percentage of times an algorithm is able to find a valid solution.

For both experiments, each planner was run 10 times since some comparative algorithms have a random nature. Thus, the quantitative results summarized in next subsections imply average and standard deviation values, while to illustrate the paths, the best one was selected for each method. An aspect to bear in mind is that for comparative algorithms, linear and angular velocities are fixed to $v = 0.5$ (m/s) and $w = -1, 1$ (rad/s), whilst for our proposal, they are adjustable.

### 3.1. Experiment 1: Navigation on an Obstacles-Free Environment

The differential robotic platform used in this study does not have inertial or global positioning sensors; in consequence, its position in the navigation environment is calculated through incremental mechanical encoders. It is well known that dead reckoning based on these sensors is subject to cumulative error. To deal with this inherent situation, the desired path is defined through a waypoint set. In this experiment, the start and goal points are

$$Start = \{[13.0, 4.75, \pi]\}, \tag{32}$$

$$Goal = \{[16.5, 17.5, 0]\}, \tag{33}$$

while the path consists of next waypoints:

$$Path = \{[13.0, 4.75, \pi]; [5.25, 8.3, \pi/2]; [5.25, 15.5, \pi/2]; [10.0, 17.5, 0]; [16.5, 17.5, 0]\}^T. \tag{34}$$

Figure 9 shows the paths generated by all evaluated algorithms during the navigation from start to goal. It can be seen that, all of them were able to reach the goal; however, it should be stressed that our proposal made the shortest route with a remarkable smoothness. These two latter aspects are of particular relevance in the mobile robotics context since they represent a lower energy consumption.

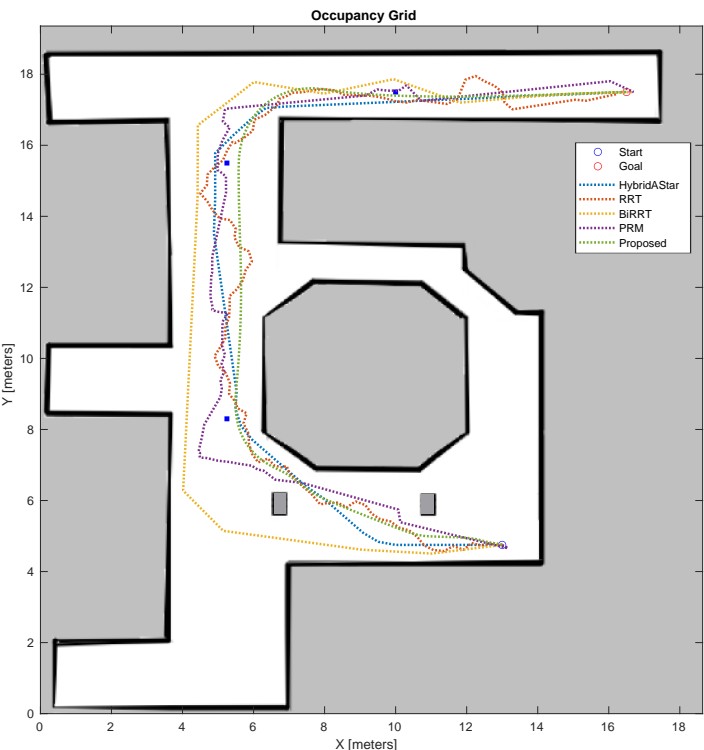

**Figure 9.** Paths obtained during the navigation on an obstacles-free environment.

Table 1 supports quantitatively previous subjective analyzing, by summarizing the average and standard deviation values. In terms of the clearance metric, the introduced approach allowed the robot to avoid the corners of the environment at a distance of 0.503 m, that is, the collision risk is lower compared to the other algorithms. For path smoothness, the current proposal achieved the lowest value, i.e., 0.236, while the RRT and PRM algorithms obtained the highest values; in other words, their paths were rougher. The BiRRT algorithm can be highlighted since it made the longest trip to reach the goal, but in the navigation task it is expected to reach the goal with less displacement. Travel time reveals that PRM, BiRRT and RRT required a shorter time than the current proposal. In addition to the aforementioned aspects, for 10 tests, all algorithms allowed the robot to reach the goal.

**Table 1.** Quantitative results for the first experiment.

| Method | Clearance (m) | Path Smoothness (rad) | Path Length (m) | Travel Time (s) | Success Rate |
|---|---|---|---|---|---|
| Hybrid AStar | $0.413 \pm 0.064$ | $0.602 \pm 0.132$ | $22.687 \pm 0.840$ | $449 \pm 41$ | 100% |
| RRT | $0.355 \pm 0.082$ | $2.336 \pm 0.615$ | $24.941 \pm 1.413$ | $418 \pm 33$ | 100% |
| BiRRT | $0.409 \pm 0.091$ | $0.826 \pm 0.265$ | $30.019 \pm 2.534$ | $396 \pm 26$ | 100% |
| PRM | $0.399 \pm 0.073$ | $1.984 \pm 0.478$ | $26.790 \pm 1.657$ | $280 \pm 19$ | 100% |
| Proposed | $0.503 \pm 0.058$ | $0.236 \pm 0.097$ | $18.289 \pm 0.283$ | $403 \pm 15$ | 100% |

In general, it can be seen that in a navigation environment free of obstacles, all the methods allowed a successful navigation of the robot. However, in real work environments, there are always obstacles.

### 3.2. Experiment 2: Navigation on a Static Obstacles Environment

The second experiment involves static obstacles placed along the path; therefore 16 were included in the environment. Figure 10 depicts how they were distributed; it also presents the trajectories traveled by the robot, when each method was evaluated. In this attempt, the start and goal points were set as

$$Start = \{[16.5, 17.5, \pi]\}, \tag{35}$$

$$Goal = \{[2.2, 1.1, \pi]\}. \tag{36}$$

The same three intermediate points were considered (with adjustments in robot orientation); in this regard, the path consists of

$$Path = \{[16.5, 17.5, \pi]; [10.0, 17.5, \pi]; [5.25, 15.5, -\pi/2]; [5.25, 8.3, -\pi/2]; [2.2, 1.1, \pi]\}^T. \tag{37}$$

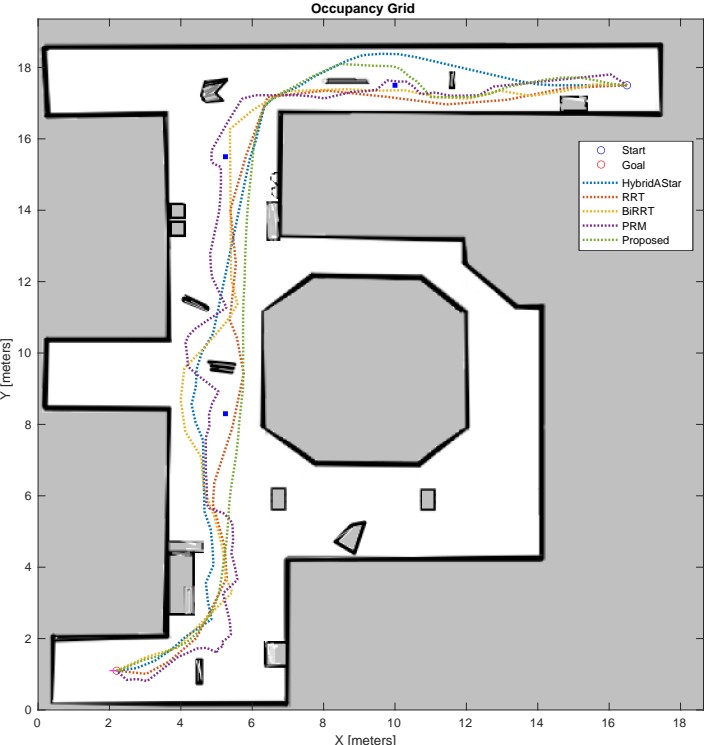

**Figure 10.** Paths obtained during the navigation on a static obstacles environment.

As can be seen in Figure 10, our proposal retains the ability to generate smoother trajectories than the comparative methods, even in the presence of obstacles. While algorithms such as RRM and PRM generated the most irregular paths, this weakness implies that for these algorithms, the robot must travel longer paths before reaching the goal. In addition, all algorithms were able to avoid the obstacles; however, for RRT, BiRRT and HybridAStar, the paths depicted a high risk of collision, in contrast to the proposed method, in which there is less problem to avoid obstacles.

Table 2 shows the overall quantitative results obtained after running each algorithm 10 times. Obstacle avoidance, which is conditional on the clearance metric, shows that algorithms such as RRT and BiRRT were more prone to object collision, and thus failed to reach the goal in some runs. The proposed method followed by HybridAStar navigated safely through the obstacles, at distances of 0.472 m and 0.390 m, respectively. Additionally,

for the path smoothness metric, the smallest value was obtained by our proposal with 0.569, followed by the HybridAstar method with 0.911. Regarding the path length that the robot had to travel before reaching the goal, it is observed that all comparative methods required an average of 2, 4, 5 and up to 7 m more than our proposal. Finally, the inclusion of obstacles reduced the efficiency of some methods. For instance, the RRT, BiRRT and PRM algorithms did not allow the robot to reach the goal in some executions, unlike the proposed method with the highest operating efficiency since it drove the robot toward the goal in all executions, although some extra time was required. Obviously, the linear and angular velocities adjustment has a direct impact on the travel time metric.

**Table 2.** Quantitative results for the second experiment.

| Method | Clearance (m) | Path Smoothness (rad) | Path Length (m) | Travel Time (s) | Success Rate |
|---|---|---|---|---|---|
| Hybrid AStar | $0.390 \pm 0.075$ | $0.911 \pm 0.242$ | $29.182 \pm 1.222$ | $538 \pm 46$ | 100% |
| RRT | $0.297 \pm 0.088$ | $2.616 \pm 0.607$ | $31.724 \pm 2.616$ | $495 \pm 37$ | 90% |
| BiRRT | $0.201 \pm 0.096$ | $2.983 \pm 0.624$ | $32.614 \pm 2.977$ | $428 \pm 28$ | 80% |
| PRM | $0.368 \pm 0.072$ | $3.421 \pm 0.712$ | $34.007 \pm 3.150$ | $345 \pm 23$ | 90% |
| Proposed | $0.472 \pm 0.047$ | $0.569 \pm 0.186$ | $27.652 \pm 0.535$ | $570 \pm 18$ | 100% |

Both experiments demonstrated that the current proposal has remarkable performance. There are, however, limitations that should be addressed to improve the efficiency. The mechanical odometry should be replaced by inertial or visual odometry to increase freedom of movement. If the granularity of the detection zones is increased, then the inputs to the fuzzy system will also increase. This condition may be untreatable since the computational cost would increase exponentially by increasing the rule set size, unless such a method as fuzzy rule interpolation is considered.

## 4. Conclusions

In this study, we submitted a reactive type-2 fuzzy approach for the real-time control of differential mobile robot navigation. The proposal was structured around three type-2 fuzzy controllers that developed behaviors for avoiding obstacles, goal reaching (implicitly, path recovering) as well as a selector between these two behaviors, according to the current condition during navigation. Through a couple of experiments, the performance of our proposal was quantitatively demonstrated, highlighting its ability to avoid obstacles while smooth paths are generated. Apart from the foregoing, this allowed reaching the goal with a much smaller displacement than the comparative methods. To demonstrate the repeatability of results to reach the goal, each of the experiments was repeated 10 times, from which we verified that our proposal was able to reach the goal with 100% effectiveness.

For future research, the inference tree will be executed in real time locally in the same robot, in such a way that it is not required to use high-performance software to perform this task. Additionally, work will be conducted to be able to navigate in environments with dynamic obstacles.

**Author Contributions:** Formal analysis, D.M.-V., A.L.-Á., A.R.-C. and J.R.; Investigation, D.M.-V., A.L.-Á., A.R.-C. and J.R.; Methodology, D.M.-V.; Software, D.M.-V., V.V.-R., A.L.-Á., A.R.-C. and M.M.-C.; Supervision, D.M.-V.; Validation, D.M.-V., V.V.-R., A.L.-Á. and M.M.C.; Writing—original draft, D.M.-V., V.V.-R., A.L.-Á., A.R.-C. and J.R.; Writing—review and editing, D.M.-V., V.V.-R., A.R.-C. and J.R. All authors have read and agreed to the published version of the manuscript.

**Funding:** This research received no external funding.

**Institutional Review Board Statement:** Not applicable.

**Informed Consent Statement:** Not applicable.

**Data Availability Statement:** Not applicable.

**Acknowledgments:** The authors thank CONACYT, as well as Tecnológico Nacional de México/Centro Nacional de Investigación y Desarrollo Tecnológico for their support.

**Conflicts of Interest:** The authors declare no conflict of interest.

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
