# Peer review of "Navigation of a Differential Wheeled Robot Based on a Type-2 Fuzzy Inference Tree"

_machines, doi:10.3390/machines10080660_

Round 1
Reviewer 1 Report
The manuscript proposes a Type-2 Fuzzy Inference Tree designed for navigation in indoor environment for a differential wheeled mobile robot. The studied topic is interesting and the manuscript has certain contributions. However, the paper can be improved as below.
1. The literature review on the approaches for obstacle avoidance in Introduction can be refined, where Dijkstra's algorithm is also popular for obstacle avoidance such as ‘Distributed multi-vehicle task assignment in a time-invariant drift field with obstacles (2019)’.
2. The main contributions of the paper are unclear. Please claim them in Introduction.
3. The grammar and styles of the manuscript need to be refined. First, English editing is required to refine sentences such as ‘which can is a very large number to be considered as possible inputs of the fuzzy controllers, for which six regions of 30◦ are considered, they are so-called Right-Down (RD), Right-Up (RU), Right-Front (RF), Left-Front(LF), Left-Up (LU) and Left-Down (LD)’, ‘The graphic 119 representation of input linguistic variables are depicted in Figure 5. RD, RU, RF, LF, LU 120 and LD are variables that allow us to quantify the proximity of obstacles, they are modeled 121 by three Gaussian type-2 fuzzy sets so called Near (N), Medium (M) and Far (F), their 122 dynamic range goes from 0.15 to 0.55 meters’, ‘Error variable is modeled by five trapezoidal 123 type-2 fuzzy sets named Negative (N), Small Negative (SN), Zero (Z), Small Positive (SP) 124 and Positive (P), this variables measures the error angle with respect to goal, its dynamic 125 range goes from −100 to 100 degrees.’, and so on. Second, a period is needed at the end of equations (6), (13), (18), (19), (20), and so on.
4. The running time of the algorithms in the two experiments is suggested to be provided for a fair comparison.
Author Response
Thank you for helping to improve the manuscript.

Reviewer 2 Report
The manuscript "Navigation of a Differential Wheeled Robot based on a Type-2 Fuzzy Inference Tree" proposes the usage of fuzzy logic to perform indoor navigation and control. The idea is interesting; however few methodological improvements can be made.
- Looking at the results observed, one concern that arises is the statistical significance of the results. How many times was each method executed? For instance, the values in tables 1, 2, and others, does represent a single execution, or are they average of how many simulations? What is the standard deviation of the results?
- The metric path smoothness from equation 30 taken from any reference? My concern is that if the method splits the turn into many small segments, the path may be very similar or worst due to the number of small segments and still be smooth for this metric. Also, It does not take into consideration the formula negative angles.
Please consider using the absolute value in the calculation.
- Is the method capable of avoiding local minimum, i.e. it can find routes in more complex environments? Has any test been made in this sense?
- Have you taken into consideration how the method would respond to problems such as dynamic maps, and the kidnapped robot problem?
- Please clarify in the manuscript if the Authors have implemented (programed) the methods used for comparison of Hybrid A* (HybridAStar), and others; or if literature implementations were used.
- If possible would be valuable to compare the performance with other works in the literature, i.e. execute the method in the same map as other published work and compare the results. For example:
https://github.com/tanujthakkar/Voronoi-Based-Hybrid-Astar
Thakkar, T., & Sinha, A. (2021, December). Motion Planning for Tractor-Trailer System. In 2021 Seventh Indian Control Conference (ICC) (pp. 93-98). IEEE.
- Moreover, some discussion about the method limitations in the results would be helpful.
Minor changes:
- There are some random words throughout the text with the first letter in caps, for example, "Type-2 Fuzzy Inference Tree". Please review it. This also applies for other acronyms such as "Same fuzzy logic controller (S.F.L.C.)".
- In some languages, the phrasing order is different from English. Thus for direct translation, the preposition "of" appears in the phrase. In the manuscript, it´s possible to notice that is being used in many locations in such manner.
For example:
"that allows monitoring the presence of nearby objects in the environment."
becomes:
"that allows nearby objects presence monitoring in the environment."
I would suggest that Authors observe this for their next manuscripts.
Author Response

(The authors gave the same response as above.)

Round 2
Reviewer 1 Report
Thanks for the efforts of the authors in refining the manuscript. Most of my comments are answered well. However, two important issues still need to be dealt with before the acceptance of the paper.
1. The literature review on shortest path planning methods in Introduction needs to be refined. Apart from the described methods, the optimal control theory based algorithm is also popular for path planning as shown in references ‘An integrated multi-population genetic algorithm for multi-vehicle task assignment in a drift field (18)’ and ‘Clustering-based algorithms for multivehicle task assignment in a time-invariant drift field (17)’.
2. In the previous comment 4, the running times of the algorithms in the two experiments mean the computational running time of the algorithms to achieve the solution, which are still missing in the updated manuscript. Please provide them with a fair comparison.
Author Response
Both comments were taken care of.

Reviewer 2 Report
After improvements and looking at the reviews that can be viewed in the current version, I believe that the work has improved. The work makes a relevant contribution to the research field and application area.
The work could benefit from more comparison with literature results; however, at this stage, the recommendation is for the article to be accepted for publication.
Author Response
We appreciate all comments given by Reviewer 2, which helped us improve the quality of this research document. Thank you for your time and orientation.
